# Low-dimensional statistical manifold embedding of directed graphs

**Thorben Funke**
L3S Research Center
Leibniz University Hannover
Hannover, Germany

**Tian Guo**
Computationl Social Science
ETH Zürich
Zurich, Switzerland

**Alen Lancic**
Faculty of Science
Department of Mathematics
University of Zagreb, Croatia

**Nino Antulov-Fantulin**
Computationl Social Science
ETH Zürich
Zurich, Switzerland
anino@ethz.ch

## Abstract

We propose a novel node embedding of directed graphs to statistical manifolds, which is based on a global minimization of pairwise relative entropy and graph geodesics in a non-linear way. Each node is encoded with a probability density function over a measurable space. Furthermore, we analyze the connection between the geometrical properties of such embedding and their efficient learning procedure. Extensive experiments show that our proposed embedding is better in preserving the global geodesic information of graphs, as well as outperforming existing embedding models on directed graphs in a variety of evaluation metrics, in an unsupervised setting.

## 1 Introduction

In this publication, we study the directed graph embedding problem in an unsupervised learning setting. A graph embedding problem is usually defined as a problem of finding a vector representation $X \in \mathbb{R}^K$ for every node of a graph $G = (V, E)$ through a mapping $\phi \colon V \to X$. On every graph $G = (V, E)$, defined with set of nodes $V$ and set of edges $E$, the distance $\mathrm{d}_G \colon V \times V \to \mathbb{R}_+$ between two vertices is defined as the number of edges connecting them in the shortest path, also called a graph geodesic. In case that $X$ is equipped with a distance metric function $\mathrm{d}_X \colon X \times X \to \mathbb{R}_+$, we can quantify the embedding distortion by measuring the ratio $\mathrm{d}_X \,/\, \mathrm{d}_G$ between pairs of corresponding embedded points and nodes. Alternatively, one can measure the quality of the mapping by using a similarity function between points. In contrast to distance, the similarity is usually a bounded value $[0, 1]$ and is in some ad-hoc way connected to a notion of distance e.g. inverse of the distance. Usually, we are interested in a low-dimensional ($K \ll n$) embedding of graphs with $n$ nodes as it is always possible to find an embedding [32] with $L_\infty$ norm with no distortion in an n-dimensional Euclidean space. Bourgain theorem (1985) [6] proved that it is possible to construct an $O(\log(n))$-dimensional Euclidean embedding of an undirected graph with $n$ nodes with finite distortion.

For the last couple of decades, different communities such as machine learning [16; 33; 10; 40; 14], physics [29; 38; 35], and computer science [42; 43] independently from mathematics community [32; 25; 12; 6] developed novel graph representation techniques. In a nutshell, they can be characterized by having one of the following properties (i-iv). **(i) The Type of loss function** that quantifies the distortion is optimizing local neighbourhood [44; 21] or global graph structure properties [53; 40; 14]. **(ii) The target property** to be preserved is either geodesic (shortest paths) [42; 43; 12] or diffusion distance (heat or random walk process) [46; 40; 14] or another similarity property [21; 55; 39]. **(iii) The mapping** $\phi$ is a linear [23; 39] or non-linear dimensionality reduction technique such as SNE [21], t-SNE [55], ISOMAP [53], Laplacian eigenmaps [3] or deep learning work DeepWalk [40], node2vec [14], HOPE [39], GraphSAGE [17], NetMF [33] and others [16]. **(iv) The geometry** of

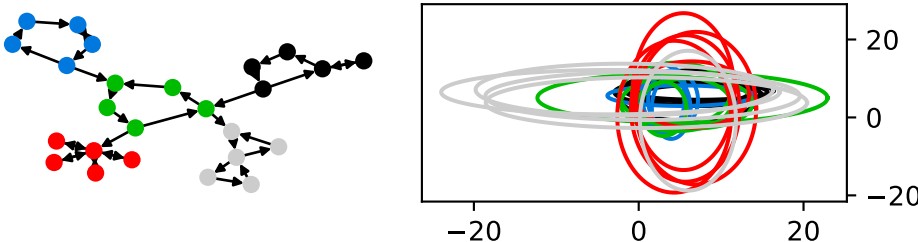

Figure 1: Visualization of synthetic example network together with our embedding. The graph is embedded into the 2-variate normal distributions, which are represented by a $\sigma$ ellipse (boundary of 1 standard deviation around mean $\boldsymbol{\mu}$). The $\sigma$ ellipses of the green nodes are contained in one of the greys nodes, which represents that the distance (measured with divergence in embedded space) between green and grey is small, but in the opposite direction very large.

$X$ has zero curvature (Euclidean) [32; 12], positive curvature [57] (spherical) or negative curvature [29; 35; 10] (hyperbolic) or mixed curvatures [15].

However, for directed graphs the asymmetry of graph shortest path distances $d_G(v_i, v_j) \neq d_G(v_j, v_i)$ is violating the symmetry assumption (not a metric function). This is the reason why only recently this problem was tackled by constructing two independent representations for each node (source and target representations) [39; 59; 27]. In this paper, we propose a single-node embedding for directed graphs to the space of probability distributions, described with the geometry of statistical manifold theory. Up to this point, different probability distributions were used for different embedding purposes such as: (i) applications to word embedding [45; 56], knowledge graphs [20], attributed graphs [4] or (ii) generalized point embeddings with elliptical distributions [36], but not for low-dimensional embedding of directed graphs, characterized by the theory of statistical manifolds.

The **main contributions** of this paper are: (i) We propose a node embedding for directed graphs to the elements of the statistical manifolds [51], to jointly capture asymmetric and infinite distances in a low-dimensional space. We determine the connection between geometry and gradient-based optimization. (ii) Furthermore, we develop a sampling-based scalable learning algorithm for large-scale networks. (iii) Extensive experiments demonstrate that the proposed embedding outperforms several directed network embedding baselines, i.e. random walks based [59] and matrix factorization based [39] methods, as well as the deep learning undirected representative [40] and deep Gaussian embeddings for attributed graphs [4].

## 2 STATISTICAL MANIFOLD DIVERGENCE EMBEDDING FOR DIRECTED GRAPHS

In this section, we analyze the limits of metric embedding. Then, we continue by providing an intuition of our embedding with a synthetic example before presenting the learning procedure of the embedding method. Bourgain theorem (1985) [6] states that it is possible to construct an $O(\log(n))$-dimensional embedding ($\phi\colon v \mapsto x_v$) of an undirected graph with $n$ nodes to Euclidean space with $O(\log(n))$ distortion by using random subset projection. But, on directed graph $G = (V, E)$, the directed shortest paths are not necessarily symmetric $d_{v,u} \neq d_{u,v}$. This does not imply that it is not possible to provide low dimensional metric embedding [6; 32; 25; 12] with a variant of Bourgain theorem. In the appendix A.1, we show that the main problem is not asymmetry, but the existence of infinite directed distances, which is quite common in real directed networks, see Table 1.

### 2.1 LEARNING STATISTICAL MANIFOLD EMBEDDING

Instead of a conventional network embedding in a metric space, we propose to use the Kullback-Leibler divergence defined on probability distributions to capture asymmetric finite and infinite distances in directed graphs. Each node is mapped to a corresponding distribution described by some parameters, which we learn during training. In this paper, we choose the class of exponential power distributions due to analytical and computational merits. With minor modifications, our method can also be applied to other distribution classes.

**Intuition.** In Fig. 1, we use a toy example to qualitatively illustrate that our embedding can represent infinite distances without requiring a high dimensional embedding space. More precisely, the network consists of five groups (color-coded), with five nodes each. The two top blocks have a connection only to the center group and, similarly, the two bottom groups have only a link from the central group to them. Our model learns this connectivity pattern by embedding members of these groups in a similar fashion. In this example, each node $u$ is embedded as a 2-variate normal distribution with mean $\boldsymbol{\mu}_u = (\mu_u^1, \mu_u^2)$ and variances $\boldsymbol{\Sigma}_u = \mathrm{diag}(\sigma_u^1, \sigma_u^2)$, which can be visualized by the $\sigma$-ellipse, the curve which represent one-$\sigma$ distance from their respective means. The nodes of graphs are embedded by minimizing the difference between the pair-wise graph distances and corresponding Kullback-Leibler divergences between embeddings. For two distribution $p(x), q(x)$, if $p(x) \gg q(x)$ on an open subset $U$, the Kullback-Leibler divergence $\mathrm{KL}(p(x), q(x))$ is then relatively high. In other words, if in Figure 1 the $\sigma$-ellipse of node $u$ is contained in or very similar to the $\sigma$-ellipse of node $v$, then the embedding represents $\mathrm{d}(u, v) < \infty$. Using this, we see that the embedding retrieved by our optimization and visualized in Fig. 1 includes most of the observed infinite distances.

**Statistical manifold embedding.** Our embedding space $X$ is the space of k-variate exponential power distributions [37] (also called generalized error distributions), described by the following probability density function for $\lambda > 0$

$$\psi_\lambda(x | \boldsymbol{\Sigma}, \boldsymbol{\mu}) = \frac{\lambda \Gamma(\frac{k}{2})}{2^{1 + \frac{k}{\lambda}} \pi^{\frac{k}{2}} \det(\boldsymbol{\Sigma})^{\frac{1}{2}} \Gamma(\frac{k}{\lambda})} \exp\left( -\frac{[(x - \boldsymbol{\mu})^T \boldsymbol{\Sigma}^{-1}(x - \boldsymbol{\mu})]^{\frac{\lambda}{2}}}{2} \right) \tag{1}$$

with mean vector $\boldsymbol{\mu} \in \mathbb{R}^k$ and covariance matrix $\boldsymbol{\Sigma} \in \mathbb{R}^{k \times k}$. $\Gamma(.)$ denotes the Gamma function and $x^T$ denotes the transposed vector of $x$. Note that $\lambda = 2$ results in the multivariate Gaussian distribution and $\lambda = 1$ yields the multivariate Laplacian distribution. As we are interested in non-degenerate distributions, we enforce positive definite co-variance matrices and further restrict ourselves to diagonal matrices, i.e. $\boldsymbol{\Sigma}_u = \mathrm{diag}(\sigma_u^1, \ldots, \sigma_u^k)$ with $\sigma_u^i \in \mathbb{R}_+$. With the latter, we reduce the degrees of freedom for each node to $2k$ and simplify our optimization by replacing a positive definite constraint on $\boldsymbol{\Sigma}_u$ with the constraints $\sigma_u^i > 0$. A common asymmetric function operating on continuous random variables is the Kullback-Leibler divergence. The asymmetric distance between nodes $u$ and $v$, denoted by $\mathrm{KL}_{u,v}$, is

$$\mathrm{KL}_{u,v} = \mathrm{KL}(p_u^\lambda, p_v^\lambda) = \int p_u^\lambda \log \frac{p_u^\lambda}{p_v^\lambda} \, \mathrm{dx}$$

with $p_u^\lambda = \psi_\lambda(x | \boldsymbol{\Sigma}_u, \boldsymbol{\mu}_u)$ and $p_v^\lambda = \psi_\lambda(x | \boldsymbol{\Sigma}_v, \boldsymbol{\mu}_v)$. We approximate it with importance sampling Monte Carlo estimation. In particular, $\mathrm{KL}_{u,v}$ can be expressed as the expectation (for $\lambda_* = 2 \leq \lambda$)

$$\mathrm{KL}_{u,v} = \mathbb{E}_{x \sim p_u^{\lambda_*}} \left[ \frac{p_u^\lambda}{p_u^{\lambda_*}} \log \frac{p_u^\lambda}{p_v^\lambda} \right],$$

where $p_u^{\lambda_*} = \psi_{\lambda_*}(x | \boldsymbol{\Sigma}_u, \boldsymbol{\mu}_u)$ is easy to sample and a proposal function for $p_\lambda(x | \boldsymbol{\Sigma}_u, \boldsymbol{\mu}_u)$. If a closed expression for $\mathrm{KL}_{u,v}$ is known, its evaluation replaces the importance sampling Monte Carlo estimation, like in the special case of $\lambda = 2$. See Appendix A.9 for more details.

Now, in order to learn these statistical manifold embeddings, we can define the loss function over asymmetric distances $D = (\mathrm{d}_{u,v})_{u,v \in V}$ of a directed graph and $\{(\boldsymbol{\mu}_u, \boldsymbol{\Sigma}_u)\}_u$ as:

$$\mathcal{L}(\{(\boldsymbol{\mu}_u, \boldsymbol{\Sigma}_u)\}_u) = \sum_{u \neq v} ||(1 + \tau \mathrm{KL}_{u,v})^{-1} - \mathrm{d}_{u,v}^{-\beta}||_2^2, \tag{2}$$

where $\tau \in \mathbb{R}_+$ is a free (trainable) parameter and $\beta \in \mathbb{R}_+$ a fixed value. This loss function given in the Eq. (2) is minimized during the learning process so that the $\mathrm{KL}_{u,v}$ based on learned node distribution representations captures the distances $\mathrm{d}_{u,v}$ in the directed graph. Empirically, we transform the given distances into finite numbers with $\mathrm{d}_{u,v}^{-\beta} \in [0, 1]$, for all $u \neq v$ and a $\beta \in \mathbb{R}_+$, which can be used to increase the differentiation between large $\mathrm{d}_{u,v}$ values and between finite and infinite distances. We modify the unbounded Kullback-Leibler divergence in a similar fashion $(1 + \tau \mathrm{KL}_{u,v})^{-1} \in [0, 1]$, where $\tau \in \mathbb{R}_+$ is a free (trainable) parameter and the additional 1 ensures a value in the same interval $[0, 1]$. We start the optimization from random initial parameters $\{\boldsymbol{\mu}_u, \boldsymbol{\Sigma}_u\}_{u \in V}$, and iteratively minimize the loss function with stochastic gradient descent optimizers, such as Adam [28]. For small $k$, which we will consider in the next section, more enhanced initialization strategies, such as using

graph plotting algorithms to determine the means and exploiting memberships of strongly connected components for the covariance initialization, probably reduce the number of training epochs.

**Scalable learning procedure.** The full objective function Eq. (2) consists of $|V|(|V|-1)$ terms and only graphs with up to the magnitude of around $10^4$ nodes can be applied. To extend our method beyond this limit, we propose an approximated solution, where the size of the training data scales linearly in the number of nodes and thus can be applied to large graphs.

This approximation solution is based on a decomposition of the loss function Eq. (2) into a neighborhood term and a singularity term as:

$$\mathcal{L}(\{(\boldsymbol{\mu}_u, \boldsymbol{\Sigma}_u)\}_u) = \underbrace{\sum_{u \neq v, \mathrm{d}_{u,v} < \infty} ||(1 + \tau \, \mathrm{KL}_{u,v})^{-1} - \mathrm{d}_{u,v}^{-\beta}||_2^2}_{\text{neighborhood term}} + \underbrace{\sum_{u \neq v, \mathrm{d}_{u,v} = \infty} ||(1 + \tau \, \mathrm{KL}_{u,v})^{-1} - \mathrm{d}_{u,v}^{-\beta}||_2^2}_{\text{singularity term}}.$$

We efficiently sample from each of these sums for each node a small number of $B$ samples and optimize our model based on the sampled information about closeness and infinite distances.

One straightforward approach to approximate the neighborhood term is to use all direct neighbors. Yet, the number of samples available via the directed neighbors is limited by $|E|$, which usually corresponds to a small $B$ in sparse real-world examples. Therefore, we apply for each node a breadth-first-search into both directions until we retrieve $B$ new samples. In this way, we obtain smaller $\mathrm{d}_{u,v}$ first, and the approximation tends to the original term in the limit $B \to |V|$.

For the singularity term, we use the topological sorting [26] of the strongly connected components. In a topological sorting of a directed acyclic graph, all edges are from lower indices to higher indices (with respect to the topological sorting).

Two important restrictions are noteworthy. First, the reverse statement does not hold, i.e. not all infinite distances are given via a single topological sorting [26]. As a consequence, we won't sample uniformly from all infinities, but from a subset of the infinities given by the topological sorting. Second, the construction of topological sorting is only possible for acyclic graphs, and most directed graphs have cycles.

Since we are only interested in sampling singularities and the graph defined by the strongly connected components of a directed graph is acyclic, we use Tarjan's algorithm [52] to retrieve the strongly connected components and topological sorting of them. After this preprocessing, we can efficiently generate samples of infinite distance for the nodes.

With these two approximations we can construct two sets $U_{\text{close}}$ and $U_\infty$ and the loss function reduces to

$$\tilde{\mathcal{L}} = \sum_{(u,v) \in U_{\text{close}}} ||(1 + \tau \, \mathrm{KL}_{u,v})^{-1} - \mathrm{d}_{u,v}^{-\beta}||_2^2 \quad + \sum_{(u,v) \in U_\infty} ||(1 + \tau \, \mathrm{KL}_{u,v})^{-1} - \mathrm{d}_{u,v}^{-\beta}||_2^2.$$

**Lemma 1.** *Let $G = (V, E)$ be a graph. Then our embedding has $2k|V| + 1$ degrees of freedom and one hyperparameter ($\beta$). Generating the training samples for the full method, is equivalent to the all pair shortest path problem, which can be solved within $O(|V|^2 \log |V| + |V||E|)$ for sparse graphs and evaluating the loss function has time complexity $O(|V|^2)$. The scalable variant has, for $B \ll |V|$, a time complexity of $O((B+1)|V| + |E|)$ and the loss function has $O(B|V|)$ terms.*

See Appendix A.2 for proof using Johnson's algorithm [24] and the pseudo-code of our methods.

## 3 STATISTICAL MANIFOLDS AND DIRECTED GRAPH GEOMETRY

In this section, we show the properties of our representation space and the effects on our learning procedure.

**Theorem 1.** *Let $\lambda$ be an even number. Then the distributions with density of Eq. (1) and parametrization $(\sigma^1, \ldots \sigma^k, \mu^1, \ldots, \mu^k) \mapsto \boldsymbol{X} \sim \psi_\lambda(x|\boldsymbol{\Sigma} = \mathrm{diag}(\sigma^1, \ldots, \sigma^k), \boldsymbol{\mu} = (\mu^1, \ldots, \mu^k))$ with $\sigma^i > 0$ are a statistical manifolds $\mathcal{S}$ with the following properties:*

*1. For univariate exponential power distribution, the curvature is constant and equal to $-1/\lambda$.*

2. *the Fisher information matrix, i.e. the Riemannian metric tensor, in this coordinate system at point $\psi_\lambda(x|\boldsymbol{\Sigma}, \boldsymbol{\mu})$ with $\boldsymbol{\Sigma} = \mathrm{diag}(\sigma^1, \ldots, \sigma^k)$ and $\boldsymbol{\mu} = (\mu^1, \ldots, \mu^k)$ is given by*

$$(g_{ij})_{ij} = \mathrm{diag}\left(\frac{c_1}{(\sigma^1)^2}, \ldots, \frac{c_1}{(\sigma^k)^2}, \frac{c_2}{(\sigma^1)^2}, \ldots, \frac{c_2}{(\sigma^k)^2}\right),$$

*where*

$$c_1 = \frac{\Gamma(1 - \frac{1}{\lambda})\lambda(\lambda - 1)}{\Gamma(\frac{1}{\lambda})}, \quad c_2 = \lambda.$$

3. *the Riemannian distance between $\psi_\lambda(x|\boldsymbol{\Sigma}, \boldsymbol{\mu})$ and $\psi_\lambda(x|\tilde{\boldsymbol{\Sigma}}, \tilde{\boldsymbol{\mu}})$ is*

$$\mathrm{d}_F(\psi_\lambda(x|\boldsymbol{\Sigma}, \boldsymbol{\mu}), \psi_\lambda(x|\tilde{\boldsymbol{\Sigma}}, \tilde{\boldsymbol{\mu}})) = \sqrt{\lambda \sum_{i=1}^{k} arcosh^2\left(1 + \frac{\left(\frac{\mu^i - \tilde{\mu}^i}{c_3}\right)^2 + (\sigma^i - \tilde{\sigma}^i)^2}{2\sigma^i \tilde{\sigma}^i}\right)}$$

*with $c_3 = \sqrt{\frac{\Gamma(\frac{1}{\lambda})}{(\lambda - 1)\Gamma(1 - \frac{1}{\lambda})}}$.*

See the Appendix A.4 for more details and proofs.

The first observation of Theorem 1 is that the geometry of the statistical manifold has constant negative curvature (see [22; 48; 58] for detailed derivation). Negative curvature also comes as a natural model for power-law degree distributions in complex networks [29; 38]. In Fig. 3 of Appendix A.5 we exemplify the negative curvature of our embedding for the graph representations of Fig. 1 by making the isometric mapping (preserving distances) to the hyperbolic space. See the Appendix A.5 for the derivation of the used isometry. Since our representation space is not flat, the Riemannian metric is different from the Euclidean. Therefore, we need to adjust the Euclidean gradient $\nabla L$ of one of our objective functions $L \in \{\mathcal{L}, \tilde{\mathcal{L}}\}$ with respect to the metric tensor [2]. The steepest descent direction is given by $\tilde{\nabla} L = G^{-1}\nabla L$, where $G^{-1}$ is the inverse of the metric tensor $G$ of $\mathcal{S}^n$ evaluated at the specific point. In other words, starting from the same representations and performing one step into the direction of $\tilde{\nabla} L$ will always result in lower values of the loss function $L$ than going into the direction of $\nabla L$. In our experiments, we report the results using $\tilde{\nabla} L$. Further discussions and details can be found in Appendix A.8.

With the last property of our embedding space, the non-Euclidean Riemannian distance, is important for embedding based applications, like clustering. Note, that the value of $c_3$ is approximately close to the value 1.0 for different values of $\lambda$. It implies that after the embedding is done with one family of distributions, we can see that the pair-wise Riemannian distance of embedded points for different representations ($\lambda$) is scaling only by the multiplicative factor $\sqrt{\lambda}$ (see Appendix A.4 for more details). Also note another useful connection, the Hessian of KL divergence is the Fisher information metric $g_{i,j}$. When the Fisher distance is small, we have $\mathrm{KL}\left(p_u^\lambda \parallel p_v^\lambda\right) \approx \frac{1}{2}\mathrm{d}_F\left(p_v^\lambda, p_u^\lambda\right)^2 \leqslant \mathrm{d}_F\left(p_v^\lambda, p_u^\lambda\right)$. Furthermore, the Fisher information metric is unique (up to scaling) and is the only Riemannian metric on the statistical manifolds [9; 8]. Due to this fact and its connection with KL divergence, this is the reason why we have not used other divergence functions.

## 4 EXPERIMENTS

### 4.1 DATASETS

From the Koblenz Network Collection [30] we retrieved three datasets of different sizes and connectivity. See Table 1 for an overview.

**Political blogs.** The small dataset is compiled during the 2004 US election [1]. In addition, we evaluate two larger networks with different proportions of reachability between their nodes.

**Cora.** [50] consists of citations between computer science publications and was used as an example in baseline APP [59] and HOPE [39] as well.

**Publication network.** With a higher density but lower reciprocity, our largest example is the publication network given by arXiv's High Energy Physics Theory (hep-th) section [31].

Table 1: Properties of datasets

| Name | $|V|$ | $|E|$ | $|\{d_{u,v} : d_{u,v} = \infty\}|/|V|^2$ | Reciprocity |
|---|---|---|---|---|
| Synthetic example | 25 | 30 | 0.48 | 34.3% |
| Political blogs | 1224 | 19,025 | 0.34 | 24.3% |
| Cora | 23,166 | 91,500 | 0.83 | 5.1% |
| arXiv hep-th | 27,770 | 352,807 | 0.71 | 0.3% |

## 4.2 BASELINES

**APP** is the asymmetric proximity preserving graph embedding method [59] based on the skip-gram model, which is used by many other methods like Node2Vec and DeepWalk. In contrast to the symmetric counterparts, APP explicitly splits the representation into a source vector and a target vector, which are updated in a direction-aware manner during the training with random walks with reset. Their method implicitly preserves the rooted PageRank score for any two vertices.

**HOPE** stands for High-Order Proximity preserved Embedding [39]. This method uses a generalization of the singular value decomposition to efficiently retrieve low-rank approximation of proximity measures like Katz Index or rooted PageRank.

**DeepWalk** is the deep learning representative of the undirected graph embedding methods. It does not differentiate between source and target and retrieves a single representation $s_u \in \mathbb{R}^K$ for each node. Like APP, DeepWalk uses the skip-gram model, trains the representation with random walks, and evaluates by cosine similarity between two node representations.

**Graph2Gauss** trains a neural network to output mean and variance of Gaussians in a way such that a ranking loss is minimized [5]. Their loss function concentrates on preserving the 1-hop and 2-hop neighborhood, but not the global graph structure ($\mathrm{d}_{u,v} > 2$). The method was designed for graphs with additional node feature data, and in the absence of such information, one-hot vector encoding should be used instead.

## 4.3 SET-UP

In this paper, we focus on directed network embedding, which preserves global properties of directed graphs. Considering this, our emphasis is different from the conventional evaluation tasks of network embedding, e.g. the link prediction task only evaluates the differentiation between $\mathrm{d}(u, v) = 1$ and $\mathrm{d}(u, v) > 1$, which respectively correspond to the case of link existence and not.

**Evaluation metrics.** Three evaluation metrics are used on the pairs of inverse true distances and approximated values derived by learned embedding from baselines and our methods. We use the *Pearson correlation coefficient* $\rho$ for checking a linear dependency [47] and *Spearman's rank correlation coefficient* $r$ for evaluating the monotonic relationship. In addition to these statistical measures, we propose to use an information-theoretic non-linear evaluation metric, i.e. *mutual information* (MI). We choose the non-parametric k-nearest-neighbor based MI estimator, LNC [13]. It is recently developed to overcome local non-uniformity and can capture relationships with limited data. We briefly describe the LNC MI here as: $\hat{I} = \frac{1}{N} \sum_{n=1} \log \frac{\hat{p}(x,y)}{\hat{p}(x) \cdot \hat{p}(y)} - \frac{1}{N} \sum_{n=1} \log \frac{\bar{V}(n)}{V(n)}$ , where $N$ is the total number of data instances, the second term is a correction term handling non-uniform region $V(n)$ surrounding point $n$, and $\hat{p}(.)$ represents the kNN density estimator.

**Hyper-parameters.** With the nonlinear interactions in our embedding, our method needs only a small number of dimensions, and for the presented results we used an embedding into a 2-variate normal distribution, i.e. $k = 2$ and the number of free parameters for each node is 4. The initial means $\boldsymbol{\mu}_u$ and co-variances $\boldsymbol{\Sigma}_u$ are randomly initialized.

For the random walk based methods APP and DeepWalk, we unified both default settings with 20 random walks for each node and length 100. The embedding dimension was set to $K = 4$, where we allowed APP to use two 4 dimensional vectors. In the same fashion, we set the embedding dimension for HOPE to 4, resulting in two $|V| \times 4$ matrices. All other parameters we left at the default value. For

our proposed embedding, we evaluate both the exact and approximate version. For the approximate one, we report the results based on $B = 10$ and $B = 100$ samples for each node.

We executed the optimization with $\beta \in \left\{ \frac{1}{4}, \frac{1}{3}, \frac{1}{2}, 1 \right\}$ and consistently retrieved the best results for $\beta = \frac{1}{2}$. For our method, we selected from the runs the point with the highest Pearson correlation between $\mathrm{d}_{u,v}^{-1}$ and $(1 + \tau \, \mathrm{KL}_{u,v})^{-1}$, which usually coincided with the minimal observed objective function value, like in Figure 1. In the experiments, we applied Adam optimizer with learning rates in $\{.001, .01, .1\}$ and retrieved the reported results with the .1 for political blogs and .001 for the others.

HOPE was executed with GNU Octave version 4.4.1, and the other methods were executed in Python 3.6.7 and Tensorflow on a server with 258 GB RAM and a NVIDIA Titan Xp GPU.

## 4.4 RESULTS

First, we compared the performance of the embedding using different $\lambda$, i.e. different classes of distributions. From the results in Appendix Table 3, we observe no significant performance improvement. So we fixed $\lambda = 2$ for the other experiments to have the performance gains of the analytical known KL divergence.

Table 2 shows the results of evaluation metrics on different datasets. $\mathrm{KL}(\cdot)$ and $\mathrm{KL}(\mathrm{full})$ refer to the approximate and exact version of our proposed embedding. We report the mean and standard deviation of MI by 40 bootstrap samples. For all metrics, the higher the value, the better the performance.

The results support this intuition that our method retrieves the highest values in all cases and significantly outperforms baselines. In addition, our approximated variant using only 10 neighbors and 10 infinities for each node already achieves higher values than baselines. Increasing the number of samples to 100, further enhances the performance of the approximate version, which even embeds the large network in a similar quality as our full method. A visualization of the results is given in Figure 2. Further details are in the Appendix A.6. Additional experiments such as dimensionality experiment, graph reconstruction, and evaluation on undirected graph can be found in Appendix A.7.

Table 2: Results including Pearson correlation coefficient $\rho$, Spearman's rank correlation coefficient $r$, and mutual information (MI). The p-value for $\rho$ and $r$ are in all cases below $10^{-8}$. For our method, we include the results with 10 samples, 100 samples for each node, and using the full distance matrix.

| Network | | APP | HOPE | DeepWalk | Graph2Gauss | KL (10) | KL (100) | KL (full) |
|---------|------|-----|------|----------|-------------|---------|----------|-----------|
| Political Blogs | $\rho$ | .16 | .45 | .25 | $-.17$ | .73 | .77 | **.88** |
| | $r$ | .29 | .45 | .24 | $-.33$ | .71 | .72 | **.89** |
| | avg. MI | .15 | .65 | .12 | .09 | .47 | .59 | **.85** |
| | std. MI | .006 | .007 | .005 | .005 | .005 | .005 | .006 |
| Cora | $\rho$ | .10 | .17 | .07 | $-.004$ | .53 | .66 | **.77** |
| | $r$ | .01 | .41 | .02 | .013 | .56 | .62 | **.65** |
| | avg. MI | .018 | .13 | .013 | .01 | .23 | .35 | **.43** |
| | std. MI | 002 | .005 | .004 | .003 | .005 | .006 | .007 |
| arXiv hep-th | $\rho$ | .01 | .20 | .08 | $-.05$ | .53 | .62 | **.68** |
| | $r$ | .12 | .28 | .04 | $-.06$ | .59 | .68 | **.72** |
| | avg. MI | .033 | .15 | .014 | .01 | .25 | **.37** | .36 |
| | std. MI | .003 | .004 | .003 | .003 | .005 | .005 | .005 |

## 5 CONCLUSION AND DISCUSSION

Although the techniques for graph embedding are quite mature [16; 33; 10; 40; 14], there still exist obstacles for using them for directed graphs. Obstacle arises from the asymmetric property of graph geodesics and a large ratio of pairs with infinite distances. Motivated by this, we propose a mapping of nodes to the elements of the statistical manifolds [51] by minimizing the divergence function between embedded points a graph geodesics. This allows a single representation that allows elegant geometrical encoding of infinite and finite distances in low-dimensional statistical manifolds by using the divergence function. One can encode an arbitrary number of infinities into the low-dimensional space, as this embedding does not need to push point on infinite geodesic distance

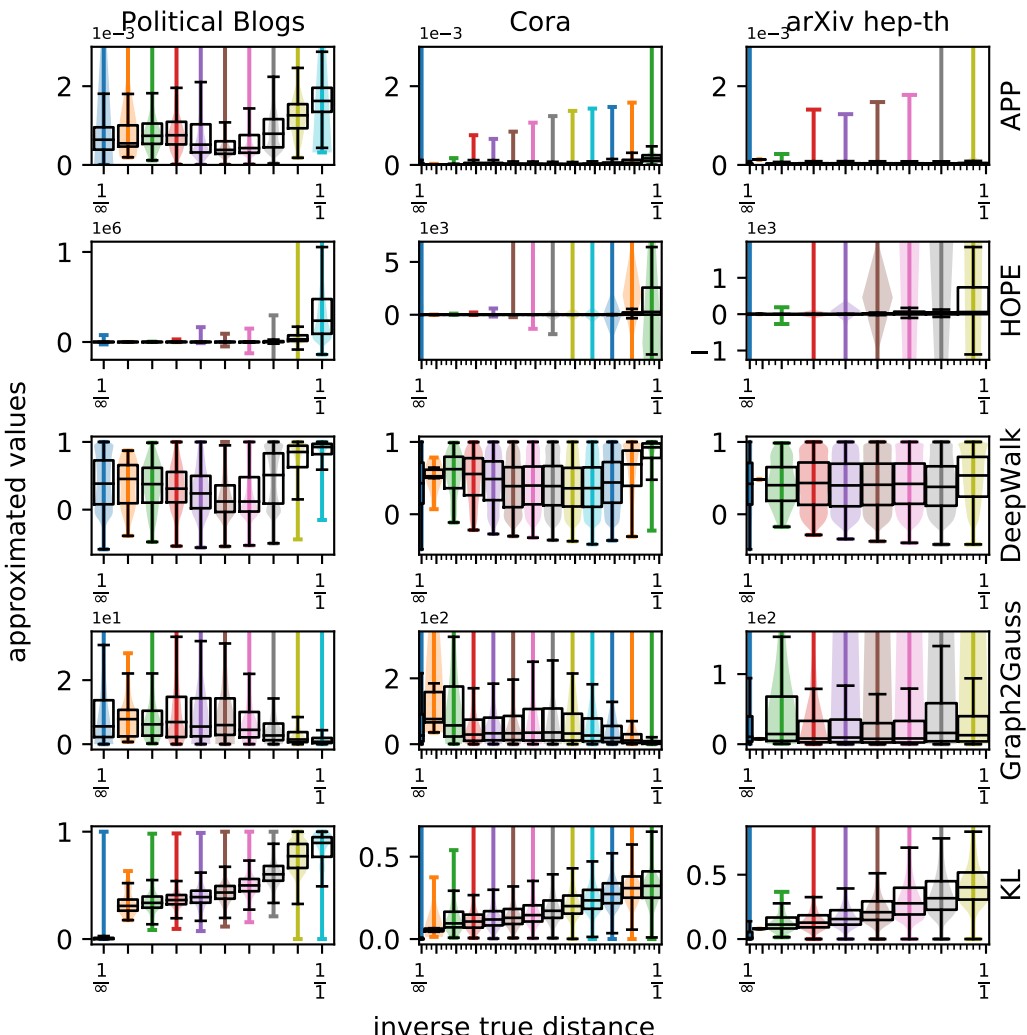

Figure 2: Visualization of approximated values w.r.t. inverse true distance $(\mathrm{d}_{u,v}^{-1})_{u,v}$ with boxplots, representing the means, first and third quartiles as well as $1.5$ interquartile range. Violin plots indicate the kernel densities. Columns respectively correspond to dataset political blogs (first column), Cora (second column), and arXiv hep-th (last column). Rows from top to bottom represent APP, HOPE, DeepWalk, Graph2Gauss, and our KL method. Ideally, as the ground-truth value increases, approximated values are expected to follow this trend as well. The baseline methods show only a weak separation between the closest distances and all other. For **our embedding method** (last row) a monotonic increasing relation is visible, which was reflected in the highest mutual information, Pearson and Spearman's correlation coefficient in Table 2.

on a manifold. Furthermore, this embedding allows the use of analytical tools from statistics and differential geometry, e.g. connection of curvature and natural gradient [2; 11; 48; 7]. In contrast to the previous work, we have characterized the geometrical structure of the underlying space to which nodes are being embedded into, and its connection to the learning via curvature corrected gradient. Furthermore, we have proposed an efficient divergence estimation method via importance sampling method. Theoretical understanding of the geometrical structures for directed graph embeddings opened many interesting theoretical and practical questions. This is the reason why we had to restrict the scope of this work only to unsupervised setting and leave link prediction and node classification task for future work.

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

## A  Appendix

### A.1  Limits of low-dimensional metric representations

**Definition 1.** We define the following metric function $\tilde{\mathrm{d}}_{u,v} = \min(\mathrm{d}_{u,v}, \mathrm{d}_{v,u})$.

**Definition 2.** For $\alpha > 1$, we say that the directed graph $G = (V, E)$ is $\alpha$-asymmetric if $\forall u, v : \mathrm{d}_{u,v} \leq \alpha \tilde{\mathrm{d}}_{u,v}$ (control of maximal asymmetry).

The following Lemma can be viewed as the corollary of Bourgain theorem. It implies that distortion w.r.t. directed graph is is not finite when $\alpha = \infty$. Note that the $\infty$-asymmetry is quite common in real directed networks, see Table 1.

**Lemma 2.** *If a graph is $\alpha$-asymmetric, then there exists a metric representations that uses random subsets embedding ($\phi \colon v \mapsto x_v$) in $m = \lceil \log(n) \rceil$ dimensional space, such that for every pair $(u, v)$ the following bounds: $||x_u - x_v||_1 / \mathrm{d}_{u,v} \leq m$ and $\mathbb{E}[||x_u - x_v||_1] / \mathrm{d}_{u,v} \geq \frac{1}{16\alpha}$ hold, where $||.||_1$ denotes the L1 norm and $\mathbb{E}[.]$ is the expectation operator over random subsets.*

*Proof.* Bourgain theorem (1985) [6] constructs an $O(\log(n))$-dimensional embedding ($\phi \colon v \mapsto x_v$) of an undirected graph with $n$ nodes to Euclidean space with $O(\log(n))$ distortion by using random subset projection. Distortion is defined as the ratio $D = \frac{\max_{u,v \in V} ||x_u - x_v||_1 / d_{u,v}}{\min_{u,v \in V} ||x_u - x_v||_1 / d_{u,v}}$, where $d_{u,v}$ denotes the shortest path in a graph and $||x_u - x_v||_1$ the L1 distance between the embedded points. For directed graphs $\mathrm{d}_{u,v}$ is not symmetric and we propose to use the metric $\tilde{\mathrm{d}}_{u,v} = \min(\mathrm{d}_{u,v}, \mathrm{d}_{v,u})$ function, which fulfills the properties of non-negativity, symmetry and triangle inequality (proof omitted).

It is possible to construct $m = \lceil \log(n) \rceil$ random subsets $A_i \subseteq V$, where each node from $V$ is put inside with the probability $1/2^i$. The following embedding for node $j$ can be constructed as $x_j = (\tilde{\mathrm{d}}(j, A_1), ..., \tilde{\mathrm{d}}(j, A_m))$, where $\tilde{\mathrm{d}}(j, A_1)$ denotes the distance from node $j$ to set $A_1$. Then, the following L1 bounds [34] for Bourgain theorem [6] for every pair $(u, v)$ hold

$$||x_u - x_v||_1 \leq m \tilde{\mathrm{d}}_{u,v}$$

and

$$\mathbb{E}[||x_u - x_v||_1] \geq \tilde{\mathrm{d}}_{u,v} / 16,$$

where $||.||_1$ denotes the L1 norm and $\mathbb{E}[.]$ the expectation operator over random subset.

Together with definition of $\tilde{\mathrm{d}}$ and $\alpha$-asymmetry gives $||x_u - x_v||_1 / \mathrm{d}_{u,v} \leq m$ and $\mathbb{E}[||x_u - x_v||_1] / \mathrm{d}_{u,v} \geq \frac{1}{16\alpha}$. It implies that distortion w.r.t. directed graph is $D \geq \frac{\max_{u,v \in V} ||x_u - x_v||_1 / d_{u,v}}{\mathbb{E}||x_u - x_v||_1 / d_{u,v}} \geq \frac{\max_{u,v \in V} ||x_u - x_v||_1 / d_{u,v}}{\frac{1}{16\alpha}}$ is not finite when $\alpha = \infty$. □

### A.2  Proof of Lemma 1

**Lemma.** *Let $G = (V, E)$ be a graph. Then our embedding has $2k|V| + 1$ degrees of freedom and one hyperparameter ($\beta$). Generating the training samples for the full method, is equivalent to the all pair shortest path problem, which can be solved within $O(|V|^2 \log |V| + |V||E|)$ for sparse graphs and evaluating the loss function has time complexity $O(|V|^2)$. The scalable variant has for $B \ll |V|$ a time complexity of $O((B + 1)|V| + |E|)$ and the loss function has $O(B|V|)$ terms.*

*Proof.* The embedding maps each node $u$ to a k-variate normal distribution $\mathcal{N}_u$ with mean $\boldsymbol{\mu}_u = (\mu_u^1, \ldots, \mu_u^k)$ and covariance $\boldsymbol{\Sigma}_u = \mathrm{diag}(\sigma_u^1, \ldots \sigma_u^k)$, which are in total $2k|V|$ parameters. With the trainable $\tau$ in our objective function, we have in total $2k|V| + 1$ degrees of freedom.

Johnson's algorithm [24] solves the problem of all pair shortest path length in $O(|V|^2 \log |V| + |V||E|)$. The result are the $|V|^2$ distances, which are used in the full loss function.

For the scalable variant, we need to perform Tarjan's algorithm [52] one time, which has time complexity of $O(|V| + |E|)$. This assigns every node its strongly connected component and at the same time returns a topological sorting of the strongly connected component. Using this, the sampling of infinities reduces to sampling from an array, which needs for $B$ samples $O(B)$. The neighborhood terms can be retrieved using a breadth-first search, which stops after finding $B$ new samples. Under the assumption of $B \ll |V|$, the total time complexity is $O((B + 1)|V| + |E|)$. Finally, the approximated loss function uses two sums over each $B|V|$ elements. □

## A.3 ALGORITHM

To clarify our two algorithms, we include the pseudo-code of the full algorithm (Algorithm 1) and the scalable variant (Algorithm 2). The main differences between Algorithm 2 and Algorithm 1 is the reduced amount of pre-processing needed and consequently the reduced amount of elements in the loss function $\tilde{\mathcal{L}}$. Lemma 1 gives the complexity for pre-processing and an optimization step.

---
**Algorithm 1** Full algorithm
---
**Input:** Graph $G = (V, E)$
1: Pre-processing: Calculate all graph distances $\mathrm{d}_{u,v}$, for all $u, v \in V$
2: Initialization: Uniformly sample random values for distribution parameters $\boldsymbol{\mu}_u, \boldsymbol{\Sigma}_u$ for each node $u \in V$
3: Optimizing: Update $\boldsymbol{\mu}_u, \boldsymbol{\Sigma}_u, \tau$ using Adam optimizer and loss function $\mathcal{L}$

---

---
**Algorithm 2** Scalable algorithm
---
**Input:** Graph $G = (V, E)$
1: Pre-processing: Calculate the strongly connected components and their topological sorting using Tarjans algorithm. Sample for each node $u \in V$ up to $B$ nodes $v$ with $\mathrm{d}_{u,v} = \infty$.
2: Pre-processing: Perform for each node $u$ a breadth-first-search for the successors and another for the predecessors to sample up to $B$ nodes with $\mathrm{d}_{u,v} < \infty$
3: Initialization: Uniformly sample random values for distribution parameters $\boldsymbol{\mu}_u, \boldsymbol{\Sigma}_u$ for each node $u \in V$
4: Optimizing: Update $\boldsymbol{\mu}_u, \boldsymbol{\Sigma}_u, \tau$ using Adam optimizer and loss function $\tilde{\mathcal{L}}$

---

## A.4 PROOF OF THEOREM 1

Let us consider a surface $S = \{\mathbf{y}(\mu, \sigma) : (\mu, \sigma) \in \mathbb{R} \times \mathbb{R}_+\}$ of natural logarithms of PDFs of univariate exponential power distributions parameterized by their expectation and deviation, $\mathbf{y}(\mu, \sigma) = \ln f(x|(\mu, \sigma)) = \ln\left(\frac{\lambda}{2\sigma\Gamma(1/\lambda)}e^{-\left(\frac{x-\mu}{\sigma}\right)^\lambda}\right)$. For given $\mu, \sigma, \lambda$ let $X$ have an exponential power distribution with those parameters and define the inner product at $(\mu, \sigma)$ as $\langle \mathbf{y}_1 | \mathbf{y}_2 \rangle (\mu, \sigma) := \mathbb{E}\left[\ln(p_1(X|\mu, \sigma)) \cdot \ln(p_2(X|\mu, \sigma))\right]$. Let us calculate the metric coefficients:

$$
\begin{aligned}
g_{11}(a\mu, \sigma) &= -\mathbb{E}\left[-\frac{a^2\lambda(\lambda-1)}{\sigma^2}\left(\frac{X-a\mu}{\sigma}\right)^{\lambda-2}\right] \\
&= \frac{a^2\lambda(\lambda-1)}{\sigma^\lambda}\mathbb{E}\left[(X-a\mu)^{\lambda-2}\right] \\
&= \frac{a^2\lambda(\lambda-1)}{\sigma^\lambda}\frac{\sigma^{\lambda-2}\Gamma\left(\frac{\lambda-1}{\lambda}\right)}{\Gamma\left(\frac{1}{\lambda}\right)} = \frac{a^2\lambda(\lambda-1)}{\sigma^2}\frac{\Gamma\left(1-\frac{1}{\lambda}\right)}{\Gamma\left(\frac{1}{\lambda}\right)},
\end{aligned}
$$

$$
g_{12}(a\mu, \sigma) = g_{21}(a\mu, \sigma) = -\mathbb{E}\left[-\frac{a\lambda^2(X-\mu)^{\lambda-1}}{\sigma^{\lambda+1}}\right] = \frac{a\lambda^2}{\sigma^{\lambda+1}}\mathbb{E}\left[(X-\mu)^{\overbrace{\lambda-1}^{\text{odd}}}\right] = 0,
$$

$$
\begin{aligned}
g_{22}(a\mu, \sigma) &= -\mathbb{E}\left[\frac{1}{\sigma^2} - \lambda(\lambda+1)\frac{(X-a\mu)^\lambda}{\sigma^{\lambda+2}}\right] \\
&= -\frac{1}{\sigma^2} + \frac{\lambda(\lambda+1)}{\sigma^{\lambda+2}}\mathbb{E}\left[(X-a\mu)^\lambda\right] = -\frac{1}{\sigma^2} + \frac{\lambda(\lambda+1)}{\sigma^2}\frac{\Gamma\left(\frac{\lambda+1}{\lambda}\right)}{\Gamma\left(\frac{1}{\lambda}\right)} = \frac{\lambda}{\sigma^2}
\end{aligned}
$$

We see that that for $a = \sqrt{\frac{\Gamma\left(\frac{1}{\lambda}\right)}{(\lambda-1)\Gamma\left(1-\frac{1}{\lambda}\right)}}$ the metric tensor at $(a\mu, \sigma)$ is given by

$$
g_F(a\mu, \sigma) = \lambda\begin{bmatrix} \sigma^{-2} & 0 \\ 0 & \sigma^{-2} \end{bmatrix} = \lambda g_{H^2}(\mu, \sigma).
$$

Similarly, for a multivariate exponential power distribution with parameters $\boldsymbol{\mu} = (\mu_1, \ldots, \mu_k), \boldsymbol{\Sigma} = \text{diag}(\sigma_1, \ldots, \sigma_k), \lambda$, we get $g_F(a\boldsymbol{\mu}, \boldsymbol{\Sigma}) = \lambda \, \text{diag}(\sigma_1^{-2}, \ldots, \sigma_k^{-2}, \sigma_1^{-2}, \ldots, \sigma_k^{-2}) = \lambda g_{H^{2k}}(\boldsymbol{\mu}, \boldsymbol{\Sigma})$ so we get

$$\|\mathbf{x}(a\boldsymbol{\mu}, \boldsymbol{\Sigma})\|_F = \sqrt{\mathbf{x}^T g_F(a\boldsymbol{\mu}, \boldsymbol{\Sigma})\mathbf{x}} = \sqrt{\mathbf{x}^T \lambda g_{H^2}(\boldsymbol{\mu}, \boldsymbol{\Sigma})\mathbf{x}} = \sqrt{\lambda}\|\mathbf{x}(\boldsymbol{\mu}, \boldsymbol{\Sigma})\|_{H^2} \Rightarrow$$

$$d_F(\mathbf{x}(a\boldsymbol{\mu_1}, \boldsymbol{\Sigma_1}), \mathbf{x}(a\boldsymbol{\mu_2}, \boldsymbol{\Sigma_2})) = \sqrt{\lambda} d_{H^{2k}}((\boldsymbol{\mu_1}, \boldsymbol{\Sigma_1}), (\boldsymbol{\mu_2}, \boldsymbol{\Sigma_2}))$$

$$= \sqrt{\lambda \sum_{i=1}^{k} \text{arcosh}^2 \left(1 + \frac{(\mu_{i1} - \mu_{i2})^2 + (\sigma_{i1} - \sigma_{i2})^2}{2\sigma_{i1}\sigma_{i2}}\right)}$$

from which it follows that

$$d_F(\mathbf{x}(\boldsymbol{\mu_1}, \boldsymbol{\Sigma_1}), \mathbf{x}(\boldsymbol{\mu_2}, \boldsymbol{\Sigma_2})) = \sqrt{\lambda \sum_{i=1}^{k} \text{arcosh}^2 \left(1 + \frac{\left(\frac{\mu_{i1} - \mu_{i2}}{a}\right)^2 + (\sigma_{i1} - \sigma_{i2})^2}{2\sigma_{i1}\sigma_{i2}}\right)}$$

The detailed derivation of the curvature for the univariate case is given in [58], where the closed form solution for $\alpha$-Gaussian curvature is given. In this paper, we are interested only in the case of $\alpha = 0$ Gaussian curvature of the Riemannian metric, which leads to the $-1/\lambda$ curvature.

## A.5 DERIVATION OF MAPPING FROM GAUSSIAN STATISTICAL MANIFOLD TO HYPERBOLOID

To see how the points on the manifold relate to each other in terms of distance, we will map them to the upper half of a two-sheathed hyperboloid (see Figure 3) while preserving their distance up to a multiplicative constant factor. To do this, we first map a point $(\mu, \theta)$ to the Poincaré half-plane through the mapping $(\mu, \theta) \mapsto (\mu/\sqrt{2}, \theta)$, which can be shown to be a similarity with the similarity coefficient $\sqrt{2}$ [7]. Then we isometrically map the Poincaré half-plane to the Poincaré disc by using the Cayley mapping [19]. We finish by mapping the Poincaré disc to the above-mentioned hyperboloid by using the inverse of the stereographic projection, which can be shown is also an isometry [49], where the distance of two points on the hyperboloid is the length of the curve which is an intersection of the hyperboloid and the plane passing through the origin and those two points. Note that the composition of these three maps is also a similarity with the similarity coefficient $\sqrt{2}$.

We have combined existing [7; 19; 49], well-known results about similarities, isometries and geodesics being mapped by those maps between the Gaussian stochastic manifold and various models for the hyperbolic geometry (Poincaré half-plane, Poincaré disc and a two-sheathed hyperboloid), using minor adjustments to suit our needs when necessary, to allow easy, direct visualisation of distances between the points on the stochastic manifold in the natural framework in which they were defined (hyperbolic geometry).

It is a well-know fact that the Cayley map $(x, y) \mapsto \left(\frac{x^2+y^2-1}{x^2+(y+1)^2}, \frac{-2x}{x^2+(y+1)^2}\right)$ is an isometry from the Poincaré half-plane to the Poincaré disc. It is also known that the stereographic projection of the upper half of the two-sheathed hyperboloid $z^2 - x^2 - y^2 = 1$ through the point $(0, 0, -1)$ to the plane $z = 0$ is an isometry from the hyperboloid onto the Poincaré disc, so it's inverse, which can be computed to be $(x, y) \mapsto \left(\frac{x}{\sqrt{1-x^2-y^2}}, \frac{y}{\sqrt{1-x^2-y^2}}, \frac{1}{\sqrt{1-x^2-y^2}}\right)$, is also an isometry. To obtain the inverse, we must find $t$ such that the point $t((x, y, 0) - (0, 0, -1)) = (tx, ty, t)$ lies on the hyperboloid, i.e. $t^2 - (tx)^2 + (ty)^2 = 1$ $\Rightarrow t = \frac{1}{\sqrt{1-x^2-y^2}}$ since we're interested only in the upper sheath. From here, we read that the inverse of the stereographic projection is as stated above.

Finally, it can be shown that a similarity $f \colon S_1 \to S_2$ maps a geodesic $\alpha$ on $S_1$ to a curve $f \circ \alpha$ on $S_2$ which can be reparameterized by arc length and such reparameterization $f \circ \alpha \circ \varphi$ is a geodesic on $S_2$. If $f$ has the similarity coefficient $c$, then it is easy to see that $d_{S_1}(x, y) = c d_{S_2}(f(x), f(y))$ by using those two geodesics and the fact that geodesics are shortest lines between points they pass through.

## A.6 FURTHER DETAILS OF EXPERIMENTS

For baseline HOPE, as input HOPE operates on any proximity measure $S$ in the form $S = M_g^{-1} M_l$, we use HOPE with $M_g = I$ the identity matrix and $M_l = ((1 - \delta_{u,v})(\text{d}_{u,v} + \varepsilon)^{-1})_{u,v}$ the matrix of element-wise inverse distances shifted by a small $\varepsilon = 10^{-6}$ to avoid division by zero. The output of HOPE are two matrices $U^s \in \mathbb{R}^{|V| \times K}, U^t \in \mathbb{R}^{K \times |V|}$ and the approximated values are calculated via $U^s \cdot U^t$.

The initial means $\mu_u^i$ are drawn uniformly from $[0, 10]$ and the initial co-variances $\sigma_u^i$ are drawn uniformly from $[4, 7]$. As initial value of $\tau$ we selected 2.5.

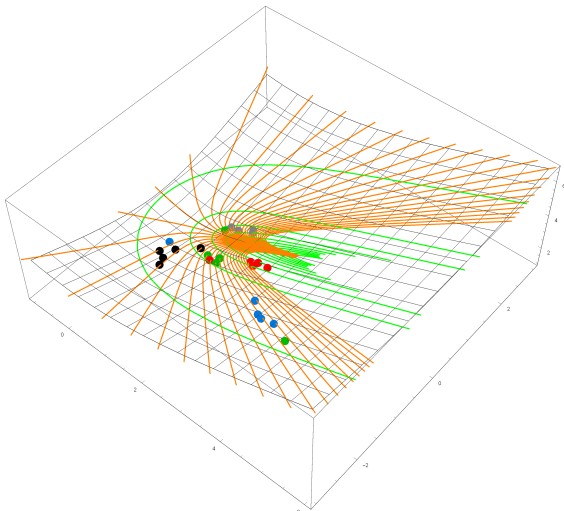

Figure 3: Visualization of the hyperboloid geometry of directed graphs for one-dimensional Gaussian distributions. The coordinate frame is shown with the green ($\sigma$) and orange lines ($\mu$). The nodes from the synthetic directed network from Figure 1 are shown as points with different colors of groups, depending on the position in the network. The embeddings for nodes ($\mu, \sigma$) in the statistical manifold are found by minimizing the objective function (2) and then making the mappings to the hyperboloid model.

Table 3: Results of the exponential power distributions for $\lambda \in \{2, 4, 8\}$ and 100 samples evaluated with Pearson correlation coefficient $\rho$, Spearman's rank correlation coefficient $r$ and mutual information (MI). $\rho$ and $r$ were evaluated on all edges $(u, v)$ with $u \neq v$ using the values given by each method and the ground truth distance. The p-value for $\rho$ and $r$ are in all cases below $10^{-8}$.

| Network | | $\lambda = 2$ | $\lambda = 4$ | $\lambda = 8$ |
|---|---|---|---|---|
| Political | $\rho$ | .77 | .78 | .78 |
| Blogs | $r$ | .74 | .74 | .72 |
| | MI | $.57 \pm .005$ | $.61 \pm .005$ | $.61 \pm .005$ |

For our full method, we used no batching for the political blogs network, 410 batches for Cora and 2777 batches for arXiv hep-th with and without shuffling between each epoch, where we saw an increased performance with shuffling especially for the larger datasets. The approximated approach uses no batch for the variant with $B = 10$ and 10 batches for $B = 100$.

The computation time of pre-processing and learning for (KL 10, KL 100, KL full) experiments were on blogs (10sec/1.5min/10min), Cora (15sec/2min/5.5hours) and arXiv hep-th (30sec/3min/1.5day).

## A.7 FURTHER EVALUATION EXPERIMENTS

In this section, we report further results of experiments with different hyperparameters, another baseline for political blogs as well as the down-stream task graph reconstruction and experiments on an undirected graph.

First, we want to verify different hyperparameters of our method. Table 3 shows the results for different exponential power distributions, i.e. $\lambda \in \{2, 4, 8\}$. The results were calculated using the proposed importance sampling, see Appendix A.9 for more details. Since for higher $\lambda$ the evaluation measures do not increase significantly, we have fixed $\lambda = 2$ for the other experiments, due to the analytical KL divergence in that case.

Another hyperparameter is the embedding dimension, which translates to selecting of k-variate exponential power distributions. Table 4 reports the results for $k \in \{2, 5, 10, 50\}$. Up to the value $k = 10$ all values increase, whereby in particular the mutual information registers the strongest increase. A further increase of the dimension to k=50 does not seem to bring any further advantage. Since the focus of this publication is on low-dimensional embeddings, we have set $k = 2$.

Table 4: Results of the k-variate exponential power distributions for $k \in \{2, 5, 10, 50\}$ of KL (full) evaluated with Pearson correlation coefficient $\rho$, Spearman's rank correlation coefficient $r$ and mutual information (MI). $\rho$ and $r$ were evaluated on all edges $(u, v)$ with $u \neq v$ using the values given by each method and the ground truth distance. The p-value for $\rho$ and $r$ are in all cases below $10^{-8}$.

| Network | | $k = 2$ | $k = 5$ | $k = 10$ | $k = 50$ |
|---|---|---|---|---|---|
| Political | $\rho$ | .88 | .90 | .91 | .88 |
| Blogs | $r$ | .89 | .90 | .92 | .90 |
| | MI | $.85 \pm .006$ | $.90 \pm .007$ | $.97 \pm .006$ | $.90 \pm .006$ |

Table 5: Comparision of results of elliptical embedding on political blogs and our method

| Method | $\rho$ | $r$ | MI |
|---|---|---|---|
| Elliptical | $-0.17$ | $-0.14$ | $.04 \pm .004$ |
| KL (full) | $0.88$ | $0.89$ | $.85 \pm .006$ |
| APP | $0.16$ | $0.29$ | $.15 \pm .006$ |
| HOPE | $0.45$ | $0.45$ | $.65 \pm .007$ |
| DeepWalk | $0.25$ | $0.24$ | $.12 \pm .005$ |
| Graph2Gauss | $-0.17$ | $-0.33$ | $.09 \pm .005$ |

For the political blogs network, we have evaluated as an additional baseline the elliptical embedding [36]. The work [36] of Muzellec et al. studies the problem of embedding objects as elliptical probability distributions, which are the generalization of Gaussian multivariate densities. Their work is rooted in the optimal transport theory by using the Wasserstein distance. The physical interpretation of this distance in the case of optimal transport is given by the cost of moving mass from one distribution to another one (see Monge–Kantorovich transportation problem [41]). In particular, for univariate Gaussian distributions, the Wasserstein distance between embedded points becomes the two-dimensional Euclidean distance of $(\mu_1, \sigma_1)$ and $(\mu_2, \sigma_2)$, i.e. flat geometry. In our case, the geometry of univariate Gaussians has constant negative curvature, and distance is measured with the Fisher distance. Our work is rooted in statistical manifold theory, where the Fisher distances arise from measuring the distinguishability between different distributions. According to the results in Table 5, we observe that this method is not outperforming our method.

Our method was created with the motivation to embed directed graphs. However, we also evaluated the performance on an undirected example. We have selected Hamsterster [18] from the Koblenz Network Collection [30]. With 2 426 nodes and 16 631 edges this friendship network has a medium size. The results in Table 6 show that even in this scenario our full method outperforms the others. Our intuition to these results is the following: Although KL divergence is an asymmetric function, in special cases it can also become symmetric. E.g. in the case of two Gaussian distributions with equal standard deviations, the KL divergence is symmetric. This demonstrates the generality of representation. Note that we did not set the additional equality constraint on the standard deviation parameters in the learning phase for this experiment. However, this only works, if all data is available and thus our approximation method does not generalize in a similar quality as seen before from the training samples to the full data set in the undirected case.

As last additional experiment, we have verified the performance of our method on the down-stream task graph reconstruction. In particular, for every node $u$ with outdegree $\theta_u$, we retrieve the $\theta_u$ best candidate successors $\mathcal{N}_{\theta_u}^{\text{out}}(u)$ from each embedding and compare them to the direct successors of node u. To assess the performance, precision is computed

$$\text{Precision}^{\text{out}} = \frac{\sum_u \mathcal{N}_{\theta_u}^{\text{out}}(u) \cap \text{Successors}(u)}{|E|}.$$

Table 6: Results on undirected network Hamsterster

| Network | | APP | HOPE | DeepWalk | Graph2Gauss | KL (10) | KL (100) | KL (full) |
|---|---|---|---|---|---|---|---|---|
| Hamsterster | $\rho$ | $-.03$ | .23 | .36 | $-.26$ | .17 | .19 | **.91** |
| | $r$ | .45 | .37 | .37 | $-.76$ | .12 | .12 | **.89** |
| | avg. MI | .29 | .43 | .10 | .45 | .63 | .64 | **.89** |
| | std. MI | .006 | .007 | .006 | .005 | .005 | .004 | .005 |

Table 7: Results of graph reconstruction for the political blogs network

| Method | out-degree precision | in-degree precision |
|---|---|---|
| APP | 0.1077 | 0.2624 |
| HOPE | 0.1010 | 0.1125 |
| DeepWalk | 0.2620 | 0.1669 |
| Graph2Gauss | 0.0258 | 0.0003 |
| Elliptical | 0.0383 | 0.0250 |
| KL (full) | 0.2861 | 0.2329 |

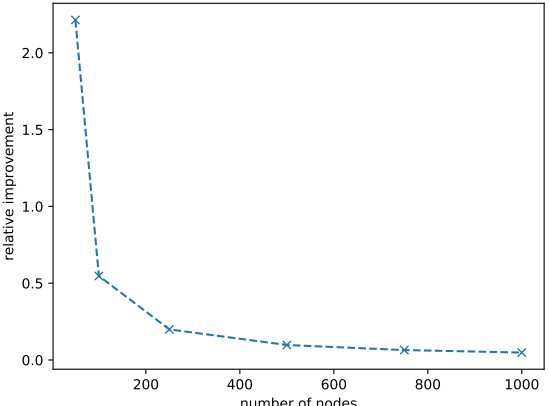

Figure 4: Relative effect of correction $\tilde{\nabla}L$ in comparison to $\nabla L$. Starting from the same representation, we optimized with gradient descent optimization with $\tilde{\nabla}L$ and $\nabla L$ with learning rate $1 * 10^{-6}$. Then the relative improvement $\frac{\Delta_{\nabla_L}L - \Delta_{\tilde{\nabla}_L}L}{\Delta_{\nabla_L}L}$ was calculated and averaged over 1000 random initialization, where $\Delta_{\nabla_L}L$ is the value of our loss function after applying gradient descent with gradient $\nabla L$. For this experimented we generated directed random networks with the Erdős–Rényi model for $p = .15$ and $n \in \{50, 100, 250, 500, 750, 1000\}$.

This tells us the ratio of actual neighboring links the embedding can extract. The same is done for incoming links, using the in-degree of each node. This kind of task was previously used in several papers that study graph embeddings e.g. [54] and [27].

Table 7 shows the resulting in- and out-precisions for the political blogs network. Although, on average, our method (KL) is performing well, it was not designed to encode only the local neighborhood but rather the whole spectrum of finite and infinite distances. Our representation is designed for preserving the global geodesic information of directed graphs in an unsupervised setting. To excel at other supervised tasks, one would have to modify the loss function to include additional terms more suitable for that down-streaming task.

### A.8 CORRECTION OF EUCLIDEAN GRADIENT

In section 3, we have deduced that the steepest direction is given by $\tilde{\nabla}L = G^{-1}\nabla L$. In other words, in each step of our optimization, we need to correct [2] the Euclidean gradient by the inverse of Fisher information matrix at the respective point. To be more specific, to update the representation of a node with $\sigma^1, \ldots \sigma^k, \mu^1, \ldots, \mu^k$ we apply

$$\tilde{\nabla}_{\frac{\partial}{\partial \sigma^i}} L = \frac{(\sigma^i)^2}{c_1} \nabla_{\frac{\partial}{\partial \sigma^i}} L, \quad \tilde{\nabla}_{\frac{\partial}{\partial \mu^i}} L = \frac{(\sigma^i)^2}{c_2} \nabla_{\frac{\partial}{\partial \mu^i}} L,$$

where $c_1$ and $c_2$ are the constants from Theorem 1. As we can see in Figure 4, the relative improvement of the corrected gradient decreases with the network size.

## A.9 IMPORTANCE SAMPLING

As there exists no closed form of KL divergence for the generalized exponential power distributions, we propose an efficient importance sampling Monte Carlo estimation, which is applicable for an even more distributions. In particular for this class of exponential power distribution, we perform

$$\mathrm{KL}_{u,v} = \mathrm{KL}(p_u^\lambda(x), q_v^\lambda(x)) = \int p_u^\lambda(x) \log \frac{p_u^\lambda(x)}{q_v^\lambda(x)} \, \mathrm{dx} = \mathbb{E}_{x \sim p_u^\lambda(x)} \left[ \log \frac{p_u^\lambda(x)}{q_v^\lambda(x)} \right].$$

However, one can also re-write the integral in order to use the easy sampling function $\psi_{\lambda_*}(x)$:

$$\int p_u^\lambda(x) \log \frac{p_u^\lambda(x)}{q_v^\lambda(x)} \, \mathrm{dx} = \int \Phi_{\lambda_*}(x) \frac{p_u^\lambda(x)}{\Phi_{\lambda_*}(x)} \log \frac{p_u^\lambda(x)}{q_v^\lambda(x)} \, \mathrm{dx} = \mathbb{E}_{x \sim \psi_{\lambda_*}(x)} \left[ \frac{p_\lambda(x)}{\psi_{\lambda_*}(x)} \log \frac{p_\lambda(x)}{q_\lambda(x)} \right].$$

For the proposal function, we will use the same class of distributions with parameter $\lambda_* = 2$. When $\lambda \geq \lambda_*$ due to the properties of exponential power distributions, the proposal distribution is concentrated around the mean from target distribution $p_u^\lambda(x)$ so the variance of estimator is relatively low. Finally, the KL divergence is approximated with $m$ samples $x_1, ...x_m$ from $\psi_{\lambda_*}(x)$:

$$\mathbb{E}_{x \sim \psi_{\lambda_*}(x)} \left[ \frac{p_\lambda(x)}{\psi_{\lambda_*}(x)} \log \frac{p_\lambda(x)}{q_\lambda(x)} \right] \approx \frac{1}{m} \sum_{i=1}^m \left[ \frac{p_\lambda(x_i)}{\psi_{\lambda_*}(x_i)} \log \frac{p_\lambda(x_i)}{q_\lambda(x_i)} \right].$$

In the special case of $\lambda = 2$ the distributions are multivariate Gaussians and the KL divergence has an analytical form. For two nodes $u, v \in V$ represented with $(\mathbf{\Sigma}_u, \boldsymbol{\mu}_u)$ respectively $(\mathbf{\Sigma}_v, \boldsymbol{\mu}_v)$ we have

$$\mathrm{KL}_{u,v} = \frac{1}{2} \left\{ \mathrm{tr}(\mathbf{\Sigma}_v^{-1} \mathbf{\Sigma}_u) + (\boldsymbol{\mu}_v - \boldsymbol{\mu}_u)^T \mathbf{\Sigma}_v^{-1} (\boldsymbol{\mu}_v - \boldsymbol{\mu}_u) - k + \ln \frac{\det \mathbf{\Sigma}_v}{\det \mathbf{\Sigma}_u} \right\}.$$

