# OpenReview forum: "Low-dimensional statistical manifold embedding of directed graphs"
_ICLR.cc/2020/Conference — Accept (Poster)_

### Official Review · AnonReviewer3 · 2019-10-18
**Official Blind Review #3**

**Rating:** 8

**Review:**

This paper proposed another graph embedding method. It focuses on directed graphs, and it embedded the graph nodes into exponential power distributions, which include the Gaussian distribution as a special case. The method is implemented by optimizing with respect to the free distributions on a statistical manifold so as to achieve the minimum distortion between the input/output distances.  The method is tested on several directed graph datasets and showed superior performance based on several metrics.

Overall, the submission forms a complete and novel contribution in the area of graph embeddings. A key novelty is that the authors used the asymmetry of KL divergences to model the asymmetry of the distances in directed graphs, and they use the fact that KL is unbounded to model the infinite distances in undirected graphs. The proposed method has three main hyperparameters, \lambda in eq.(1), \beta in eq.(2), and the dimensionality of the target embedding. The author showed that \lambda and \beta are not sensitive and can be set to the default values, and 2-dimensional distributions already give much better results as compared to alternative embeddings. Moreover, the author proposed a scalable implementation based on sampling. Furthermore, the authored justified their choice of the target embedding space through some minor theoretical analysis given in section 3.

The writing quality and clarity are good (well above average).

To further improve this paper (e.g., in the final version), the authors are suggested to incorporate the following comments:

In the experimental evaluation, it should include some cases when the dimensionality of the target embedding has a large value (e.g., 50). This will make the evaluation more complete.

There are some typos and unusual expressions.  For example, page 3, what is "a good proposal function"?

After eq.(1), mention \lambda is a hyperparameter (that is not to be learned).

Theorem 1 (2), mention the Fisher information matrix is wrt the coordinate system (\sigma^1, \cdots,\sigma^k, \mu^1, \cdots, \mu^k)

Ideally, the experiments can include an undirected graph and show for example that the advantages of the proposed method become smaller in this case.

**Experience Assessment:**

I have published in this field for several years.

**Review Assessment: Checking Correctness Of Derivations And Theory:**

I carefully checked the derivations and theory.

**Review Assessment: Checking Correctness Of Experiments:**

I assessed the sensibility of the experiments.

**Review Assessment: Thoroughness In Paper Reading:**

I read the paper thoroughly.

---

> ### Author Response · Authors · 2019-11-13
> **Response to Reviewer #3**
>
> We include the results where we use 5-variate, 10-variate, and 50-variate exponential power distribution. Note that the k-variate exponential power distribution is parametrized by a mean vector (k-dimensional) and diagonal covariance matrix with k parameters.
> In particular, for the Political Blogs, with KL Full on Political blogs we obtain the following:
>
> distribution dimensionality (Pearson, Spearman, avg. MI, std. MI)
> 2-variate   0.88, 0.89, 0.85, 0.006
> 5-variate   0.90, 0.90, 0.90, 0.007
> 10-variate 0.91, 0.92, 0.97, 0.006
> 50-variate 0.88, 0.90, 0.90, 0.006.
>
> We observe that low dimensional embeddings (2-variate distributions) are quite efficient w.r.t. different performance measures.
> We have changed the phrase "good proposal function", and we have included more details around this in Appendix A.8. Additionally, we have corrected the wordings and fixed the typos.
>
> As suggested by the reviewer, we have made additional experiments on the undirected network.
> Results on Petster-hamster network (http://konect.uni-koblenz.de/networks/petster-hamster)
> Method 		(Pearson, Spearman, avg. MI, std. MI):
> APP 		 (-0.03, 0.45, 0.29, 0.006)
> HOPE 		 (0.23, 0.37, 0.43, 0.007)
> DeepWalk 	 (0.36, 0.37, 0.10, 0.006)
> Graph2Gauss (-0.26, -0.76, 0.45, 0.005)
> KL 		          (0.91 , 0.89, 0.89, 0.005)
>
> We observe that our representation is still capable of preserving the global geodesic information of undirected graphs. Why is that the case if undirected graphs have symmetric shortest path distances between nodes?
> Although KL divergence is an asymmetric function, in special cases it can also become symmetric. E.g. in the case of two Gaussian distributions with equal standard deviations, the KL divergence is symmetric. This demonstrates the generality of representation.  Note that we did not set the additional equality constraint on the standard deviation parameters in the learning phase for this experiment.
>
> An updated version of the paper will be uploaded soon.

---

> > ### Comment · AnonReviewer3 · 2019-11-13
> > **Acknowledging Rebuttals**
> >
> > The current reviewer has read the authors' rebuttal.
> >
> > Based on my comments, the author performed additional experiments including (1) increased dimensionality of the target embedding; (2) embedding an undirected graph. These additional experiments have further strengthened this contribution.

---

### Official Review · AnonReviewer1 · 2019-10-22
**Official Blind Review #1**

**Rating:** 6

**Review:**

In this paper, the authors proposed an embedding method for directed graphs.
Each node is represented by a normal distribution.
Accordingly, for each pair of nodes, the authors used the KL divergence between their distributions to fit the observed distance.
The asymmetric property of the KL divergence matches well with the nature of directed graphs.
A scalable algorithm is designed.
The property of the learned space is analyzed in detail, which verifies the rationality of using KL divergence.

My main concerns are about the experiments and the baselines.

1. The baselines are not very representative. Except for APP, the remaining three methods are not motivated by directed graphs. To my knowledge, authors can further consider the following two methods as their baselines.

a) The classic method like the Okada-Imaizumi Radius Distance Model in “Okada, Akinori, and Tadashi Imaizumi. "Nonmetric multidimensional scaling of asymmetric proximities." Behaviormetrika 14.21 (1987): 81-96.” This method represents each node in a directed graph as an embedding vector with a radius and proposed a Hausdorff-like distance.

b) The recent work in [35]. This method also embeds nodes by elliptical distributions, but the distance is measured in the Wasserstein space.

This work will be stronger if the authors discuss the advantages of the proposed model compared with these two methods and add more comparison experiments.


2. In practice, node embeddings are always used in down-stream learning tasks. Besides the classic statistical measurements, I would like to see a down-stream application of the proposed method, e.g., node classification/clustering. Adding such an experiment will make the advantage of the proposed method more convincing.

**Experience Assessment:**

I have published one or two papers in this area.

**Review Assessment: Checking Correctness Of Derivations And Theory:**

I assessed the sensibility of the derivations and theory.

**Review Assessment: Checking Correctness Of Experiments:**

I carefully checked the experiments.

**Review Assessment: Thoroughness In Paper Reading:**

I read the paper at least twice and used my best judgement in assessing the paper.

---

> ### Author Response · Authors · 2019-11-13
> **Response to Reviewer #1**
>
>
> 1. In our paper, we have used the following baselines for directed graphs: HOPE [Ou et al. in ACM SIGKDD 2016], APP [ZHOU et al. in AAAI 2017] and Graph2Gauss [BOJCHEVSKI et al. in ICLR 2018]. All of them (HOPE, APP, Graph2Gauss) have explicitly written that they consider directions. We will make sure to stress this in our paper. However, DeppWalk model [Perozzi et. al. in ACM SIGKDD 2014] was only used as a representative baseline for the un-directed class of algorithms.
>
> We thank the reviewer for the suggestions for the new baselines. However, Okada-Imaizumi model [Behaviormetrika 14.21 (1987)] was under a pay-wall that was not accessible with our institution's subscription. Without any success, we have tried our best to find any publicly available material on the manuscript or the code.
> Furthermore, as you have suggested, the work MUZELLEC et al. in NIPS 2018 [35] is available and we have included additional experiments with elliptical embedding. We observe that elliptical embedding is not outperforming our method on Political Blogs network:
> method      (Pearson, Spearman, avg. MI, std. MI)
> KL                      (0.88,  0.89, 0.85, 0.006)
> Elliptical           (-0.17, -0.14, 0.036, 0.004)
> APP                   (0.16,  0.29, 0.15, 0.006)
> HOPE                (0.45,  0.45, 0.65, 0.007)
> Graph2Gauss  (-0.17, -0.33, 0.09, 0.005)
> DeepWalk         (0.25,  0.24, 0.12, 0.005)
>
> The work of Muzellec et al. studies the problem of embedding objects as elliptical probability distributions, which are the generalization of Gaussian multivariate densities. Their work is rooted in the optimal transport theory by using the Wasserstein distance. The physical interpretation of this distance in the case of optimal transport is given by the cost of moving mass from one distribution to another one. In particular, for univariate Gaussian distributions, the Wasserstein distance between embedded points becomes the two-dimensional Euclidean distance of ($\mu_1,\sigma_1$) and ($\mu_2,\sigma_2$), i.e. flat geometry.
> In our case, the geometry of univariate Gaussians has constant negative curvature, and distance is measured with the Fisher distance. Our work is rooted in statistical manifold theory, where the Fisher distances arise from measuring the distinguishability between different distributions.
>
> 2. Since the down-stream application was also raised by reviewer #2 point 2, we reply in a joint manner:
> We agree that the majority of node embeddings are used for down-stream learning tasks. In this paper, we have focused on the unsupervised setting and finding representations that can preserve geodesic relationships between nodes in a directed graph first. We have focused on the graph representation itself, its geometrical properties, connections to existing mathematical frameworks and learning. We believe that for supervised tasks a modification to the loss function is needed.
> At the same time, we agree with the reviewer and thus include a down-stream learning task that is suitable for the unsupervised scenarios. In particular, for every node u with outdegree k, we retrieve the k best candidate neighbors from each embedding and compare them to the direct successors of node u. To assess the performance, precision is computed, which tells us the ratio of actual neighboring links the embedding can extract. The same is done for incoming links, using the in-degree of each node. This kind of task was previously used in several papers that study graph embeddings e.g. [Tsitsulin et al., WWW 2018] and [Khosla et al., ECMLPKDD 2019].
> Here, we report the precision for outdegree and indegree link reconstruction for Political Blogs network.
> Method 		        (outdegree precision, indegree precision)
> APP 		        (0.1077, 0.2624)
> HOPE 		        (0.1010, 0.1125)
> DeepWalk 	        (0.2620, 0.1669)
> Graph2Gauss 	(0.0258, 0.0003)
> Elliptical 		(0.0383, 0.0250)
> KL 		                (0.2861, 0.2329)
>
>
> Although, on average, our method (KL) is performing well, it was not designed to encode only the local neighborhood but rather the whole spectrum of finite and infinite distances. In our paper, we claim that our representation is designed for preserving the global geodesic information of directed graphs in an unsupervised setting. To excel at other supervised tasks, one would have to modify the loss function to include additional terms more suitable for that down-streaming task, which is part of our future work.
>
>
> An updated version of the paper is uploaded.

---

### Official Review · AnonReviewer2 · 2019-10-23
**Official Blind Review #2**

**Rating:** 6

**Review:**

This paper proposes an unsupervised method for learning node embeddings of directed graphs into statistical manifolds. Each node in the graph is mapped to a distribution in the space of k-variate power distributions, endowed with the KL divergence as asymetric similarity. The authors propose an optimization method based on a regularized KL divergence objective. They also propose an approximation of this objective based on finite neighborhoods, with a separate treatment of infinite distances based on a topological sorting. They also introduce a natural gradient correction to the gradient descent algorithm in this setting. They validate the fitness of their approach by showing that asymmetric distances in the graph translate into correlated asymetric distances between the node embeddings for various datasets.

The paper appears to bring a valuable method for directed graph embedding. However, a more thorough experimental study would help validating the improvements and hyperparameter setting of the method. Moreover, I suggest that the authors work on an improved version of the manuscript, as it contains many grammatical and spelling mistakes, some of which listed under.

* Experimental questions *
1. The hyperparameters and training time seems to have been set using the evaluation metric on the datasets. Could the authors provide a more principled validation approach to their experiments, e.g. using cross-validation?
2. While the focus on preserving asymetric similarities is understandable, it would be interesting to know how the method performs for conventional evaluation tasks of network embedding, and to show that the gain in correlation can translate into gains for the end task in practice.

* Spelling / grammar / layout *
- Title: “On the geometry and learning low-dimensional embeddings…” does not make sense.
- abstract: “is better preserving the global geodesic”
- Fig 1: The sigma ellipse*s*
- 2.1 Intuition.: “color codded”
- Figure 2: “which was reflected the highest mutual information”
- Figure 2 should visually identify the rows and the columns, rather than relying on the caption.


**Experience Assessment:**

I do not know much about this area.

**Review Assessment: Checking Correctness Of Derivations And Theory:**

I assessed the sensibility of the derivations and theory.

**Review Assessment: Checking Correctness Of Experiments:**

I assessed the sensibility of the experiments.

**Review Assessment: Thoroughness In Paper Reading:**

I read the paper thoroughly.

---

> ### Author Response · Authors · 2019-11-13
> **Response to Reviewer #2**
>
>
> * Experimental questions *
> 1.  In the updated version, we correct grammatical mistakes and polish the content that was not clear. Cross-validation was not used since an unsupervised setting is used. The training time or, more specifically, the number of training epochs were selected with the convergence criteria of the loss function. We will extend the Appendix A.6, to provide more details on the experimental procedure and hyperparameters (lambda - distribution class, beta - distance scaling exponent, learning rate in ADAM optimizer and batch size).
>
> 2. Since the down-stream application was also raised by reviewer #1 point 2, we reply in a joint manner:
> We agree that the majority of node embeddings are used for down-stream learning tasks. In this paper, we have focused on the unsupervised setting and finding representations that can preserve geodesic relationships between nodes in a directed graph first. We have focused on the graph representation itself, its geometrical properties, connections to existing mathematical frameworks and learning. We believe that for supervised tasks a modification to the loss function is needed.
> At the same time, we agree with the reviewer and thus include a down-stream learning task that is suitable for the unsupervised scenarios. In particular, for every node u with outdegree k, we retrieve the k best candidate neighbors from each embedding and compare them to the direct successors of node u. To assess the performance, precision is computed, which tells us the ratio of actual neighboring links the embedding can extract. The same is done for incoming links, using the in-degree of each node. This kind of task was previously used in several papers that study graph embeddings e.g. [Tsitsulin et al., WWW 2018] and [Khosla et al., ECMLPKDD 2019].
> Here, we report the precision for outdegree and indegree link reconstruction for Political Blogs network.
> Method 		        (outdegree precision, indegree precision)
> APP 		        (0.1077, 0.2624)
> HOPE 		        (0.1010, 0.1125)
> DeepWalk 	        (0.2620, 0.1669)
> Graph2Gauss 	(0.0258, 0.0003)
> Elliptical 		(0.0383, 0.0250)
> KL 		                (0.2861, 0.2329)
>
> Although, on average, our method (KL) is performing well, it was not designed to encode only the local neighborhood but rather the whole spectrum of finite and infinite distances. In our paper, we claim that our representation is designed for preserving the global geodesic information of directed graphs in an unsupervised setting. To excel at other supervised tasks, one would have to modify the loss function to include additional terms more suitable for that down-streaming task, which is part of our future work.
>
>
> * Spelling / grammar / layout *
> We revised all the writing issues and have rephrased the title to be more clear.
>
> An updated version of the paper will be uploaded soon.

---

### Decision · Program_Chairs · 2019-12-19

**Decision:**

Accept (Poster)

**Comment:**

The paper proposes an embedding for nodes in a directed graph, which takes into account the asymmetry. The proposed method learns an embedding of a node as an exponential distribution (e.g. Gaussian), on a statistical manifold. The authors also provide an approximation for large graphs, and show that the method performs well in empirical comparisons.

The authors were very responsive in the discussion phase, providing new experiments in response to the reviews. This is a nice example where a good paper is improved by several extra suggestions by reviewers. I encourage the authors to provide all the software for reproducing their work in the final version.

Overall, this is a great paper which proposes a new graph embedding approach that is scalable and provides nice empirical results.